# Not All Causal Inference is the Same

**Matej Zečević**                                             *matej.zecevic@tu-darmstadt.de*
*Computer Science Department, TU Darmstadt, Germany*

**Devendra Singh Dhami**                                      *devendra.dhami@tu-darmstadt.de*
*Computer Science Department, TU Darmstadt, Germany*
*Hessian Center for AI (hessian.AI), Germany*

**Kristian Kersting**                                         *kersting@cs.tu-darmstadt.de*
*Computer Science Department, TU Darmstadt, Germany*
*Centre for Cognitive Science, TU Darmstadt, Germany*
*Hessian Center for AI (hessian.AI), Germany*
*German Research Center for Artificial Intelligence (DFKI), Germany*

**Reviewed on OpenReview:** *https: // openreview. net/ forum? id= ySWQ6eXAKp*

## Abstract

Neurally-parameterized Structural Causal Models in the Pearlian notion to causality, referred to as NCM, were recently introduced as a step towards next-generation learning systems. However, said NCM are only concerned with the learning aspect of causal inference and totally miss out on the architecture aspect. That is, actual causal inference within NCM is intractable in that the NCM won't return an answer to a query in polynomial time. This insight follows as corollary to the more general statement on the intractability of arbitrary structural causal model (SCM) parameterizations, which we prove in this work through classical 3-SAT reduction. Since future learning algorithms will be required to deal with both high dimensional data and highly complex mechanisms governing the data, we ultimately believe work on tractable inference for causality to be decisive. We also show that not all "causal" models are created equal. More specifically, there are models capable of answering causal queries that are not SCM, which we refer to as *partially causal models* (PCM). We provide a tabular taxonomy in terms of tractability properties for all of the different model families, namely correlation-based, PCM and SCM. To conclude our work, we also provide some initial ideas on how to overcome parts of the intractability of causal inference with SCM by showing an example of how parameterizing an SCM with SPN modules can at least allow for tractable mechanisms.

With this work we hope that our insights can raise awareness for this novel research direction since achieving success with causality in real world downstream tasks will not only depend on learning correct models but also require having the practical ability to gain access to model inferences.

## 1 Introduction

Causal interactions stand at the center of human cognition thus being of high value to science, engineering, business, and law (Penn & Povinelli, 2007). Questions like "What if?" and "Why?" were discovered to be central to how children perform exploration, as recent strides in developmental psychology suggest (Gopnik, 2012; Buchsbaum et al., 2012; Pearl & Mackenzie, 2018), and similar to the scientific method. Whereas artificial intelligence research dreams of an automatation to the scientist's manner (McCarthy, 1998; McCarthy & Hayes, 1981; Steinruecken et al., 2019). Deep learning's advance brought universality in approximation i.e., for any function there will exist a neural network that is close in approximation to arbitrary precision (Cybenko, 1989; Hornik, 1991). The field has seen tremendous progress ever since, see

for instance (Krizhevsky et al., 2012; Mnih et al., 2013; Vaswani et al., 2017). Thereby, the integration of causality with deep learning is crucial for achieving human-level intelligence. Preliminary attempts, for the so-called neural-causal models (Xia et al., 2021; Pawlowski et al., 2020; Zečević et al., 2021a) suggest to be a promising step in this direction.

While causality has been thoroughly formalized within the last decade (Pearl, 2009; Peters et al., 2017), and deep learning has advanced at a breakneck speed, the issue of tractability of inference (Cooper, 1990; Roth, 1996; Choi et al., 2020) has been left relatively unscathed. It is generally known that semantic graphs like Bayesian Networks (BNs, Pearl (1995)) scale exponentially for marginal inference, while computation graphs (or probabilistic circuits) like sum-product networks (SPNs, Poon & Domingos (2011)) scale in polynomial (if not linear) time. A conversion method developed by Zhao et al. (2015) showed how to compile back and forth between SPNs and BNs. Yet, diverging views on tractable causal inference were reported, as discussed in Papantonis & Belle (2020) and Zečević et al. (2021a). The former argues using the aforementioned conversion scheme, which leads to a degenerate BN with no causal semantics, while the latter proposes a partial neural-causal model that leverages existing interventional data to perform tractable causal inferences. Motivated by these discrepancies and the resulting lack of clarity, this work focuses on investigating systematically if, when, how and also under what cost the different types of causal inference occur in tractable manner. Our investigation of causal models reveals a tabular taxonomy to summarize recent research efforts and clarifies what should be subject of further research. Going a step further we also reveal a newly proposed model, *Linear Time Neural Causal Model* (LTNCM) as an initial step towards the said goal.

We make the following contributions: (1) We prove the general impossibility result of tractable inference within parameterized SCM, (2) we identify the differences in existing causal models out of which we arrive at and define the new class of *partially causal models*, (3) we provide a comprehensive view onto the different trade-offs between model expressivity and inference tractability, classifying our models along with their properties within a tabular taxonomy, and finally (4) based on our taxonomy we propose a new model called LTNCM that can perform linear time mechanism inference opposed to polynomiality of a standard NCM.

We make our code repository for reproducing the empirical part with the LTNCM and visualizations publicly available at: `https://github.com/zecevic-matej/Not-All-Causal-Inference-is-the-Same`

## 2    Brief Overview on Background and Related Work

Let us briefly review the background on both key concepts from causality and the main tractable model class of concern, sum-product networks (SPNs). Because SPNs will play a central role in the discussion of this paper, since they take a singular role in model families that are truly tractable, we refer readers unfamiliar with the model family to the overview provided by París et al. (2020).

**Causal Inference.** Following the Pearlian notion of Causality (Pearl, 2009), an SCM is defined as a 4-tuple $\mathcal{M} := \langle \mathbf{U}, \mathbf{V}, \mathcal{F}, P(\mathbf{U}) \rangle$ where the so-called structural equations

$$v_i \leftarrow f_i(\mathrm{pa}_i, u_i) \in \mathcal{F} \tag{1}$$

assign values (denoted by lowercase letters) to the respective endogenous/system variables $V_i \in \mathbf{V}$ based on the values of their parents $\mathrm{Pa}_i \subseteq \mathbf{V} \backslash V_i$ and the values of their respective exogenous/noise/nature variables $\mathbf{U}_i \subseteq \mathbf{U}$, and $P(\mathbf{U})$ denotes the probability function defined over $\mathbf{U}$. An intervention $do(\mathbf{W}), \mathbf{W} \subset \mathbf{V}$ on an SCM $\mathcal{M}$ occurs when (multiple) structural equations are being replaced through new non-parametric functions thus effectively creating an alternate SCM. Interventions are referred to as *imperfect* if the parental relation is kept intact, as *perfect* if not, and even *atomic* when additionally the intervened values are being kept constant. It is important to realize that interventions are of fundamentally *local* nature, and the structural equations (variables and their causes) dictate this locality. This further suggests that mechanisms remain invariant to changes in other mechanisms. An important consequence of said autonomic principles is the *truncated factorization*

$$p^{\mathcal{M}}(\mathbf{v}) = \prod\nolimits_{V_i \notin \mathbf{W}} p(v_i \mid \mathrm{pa}_i) \tag{2}$$

derived by Pearl (2009), which suggests that an intervention $do(\mathbf{W})$ introduces an independence of a set of intervened nodes $\mathbf{W}$ to its causal parents. For completion we mention more interesting properties of any SCM,

they induce a causal graph $G$ as directed acyclic graph (DAG), they induce an observational/associational distribution denoted $p^{\mathcal{M}}$, and they can generate infinitely many interventional and counterfactual distributions using the *do*-operator which "overwrites" structural equations. Note that, opposed to the Markovian SCM discussed in for instance (Peters et al., 2017), the definition of $\mathcal{M}$ is semi-Markovian thus allowing for shared $U$ between the different $V_i$. Such a $U$ is also called *hidden confounder* since it is a common cause of at least two $V_i, V_j (i \neq j)$. Opposite to that, a "common" confounder would be a common cause from within $\mathbf{V}$. The SCM's applicability to machine learning has been shown in marketing (Hair Jr & Sarstedt, 2021), healthcare (Bica et al., 2020) and education (Hoiles & Schaar, 2016). As suggested by the Causal Hierarchy Theorem (CHT) (Bareinboim et al., 2020), the properties of an SCM form the Pearl Causal Hierarchy (PCH) consisting of different levels of distributions being $\mathcal{L}_1$ *associational*, $\mathcal{L}_2$ *interventional* and $\mathcal{L}_3$ *counterfactual*. The PCH suggests that causal quantities ($\mathcal{L}_i, i \in \{2, 3\}$) are in fact richer in information than statistical quantities ($\mathcal{L}_1$), and the necessity of causal information (e.g. structural knowledge) for inference based on lower rungs e.g. $\mathcal{L}_1 \nrightarrow \mathcal{L}_2$. Finally, to query for samples of a given SCM, the structural equations are being simulated sequentially following the underlying causal structure starting from independent, exogenous variables $U_i$ and then moving along the causal hierarchy of endogenous variables $\mathbf{V}$.

**Sum-Product Networks.** To readers unfamiliar with SPN literature, the present paragraph should serve as a general, high-level introduction to the topic. Note however that this paragraph should not be considered as an in-depth dive/tutorial into the topic, for further consideration please do consider the provided survey and original references. We cover the basics only up to the point that they are relevant for this present manuscript. Wile some other details like the definition of a 'scope' or 'indicator variables' are integral for actually implementing SPNs and SPN-based causal models, the reader is safe to ignore those since they won't be relevant for the (technical) arguments given in this paper. We follow suit with existing literature and the recent strides on tractable causal inference—mainly revolving around sum-product networks (SPN) as introduced by Poon & Domingos (2011). SPNs generalized the notion of network polynomials based on indicator variables $\lambda_{X=x}(\mathbf{x}) \in \{0, 1\}$ for (finite-state) RVs $\mathbf{X}$ from (Darwiche, 2003). The indicator variable (IV) $\lambda$ simply denotes whether a certain state $x$ is present for a random variable (RV) $X$ that is part of a collection of RVs denoted $\mathbf{X}$ and if so, then $\lambda_{X=x}(\mathbf{x}) = 1$. Sum-product networks (SPN) represent a special type of probabilistic model that allows for a variety of exact and efficient inference routines. SPNs are considered as DAG consisting of product, sum and leaf (or distribution) nodes whose structure and parameterization can be efficiently learned from data to allow for efficient modelling of joint probability distributions $p(\mathbf{X})$. Formally a SPN $\mathcal{S} = (G, \mathbf{w})$ consists of non-negative parameters $\mathbf{w}$ and a DAG $G = (V, E)$ with a multivariate indicator variable $\boldsymbol{\lambda}$ leaf nodes and exclusively internal sum (denoted S) and product nodes (denoted P) given by,

$$\mathsf{S}(\boldsymbol{\lambda}) = \sum_{\mathsf{C} \in \mathrm{ch}(\mathsf{S})} \mathbf{w}_{\mathsf{S},\mathsf{C}} \mathsf{C}(\boldsymbol{\lambda}) \quad \mathsf{P}(\boldsymbol{\lambda}) = \prod_{\mathsf{C} \in \mathrm{ch}(\mathsf{S})} \mathsf{C}(\boldsymbol{\lambda}), \tag{3}$$

where $\mathsf{S}(\boldsymbol{\lambda})$ and $\mathsf{P}(\boldsymbol{\lambda})$ denote the evaluation of sum and product nodes using the leaf node ($\boldsymbol{\lambda}$) propagation respectively, and each of these nodes is being computed recursively based on their children denoted by $\mathrm{ch}(\cdot)$, and $\mathbf{w}_{\mathsf{S},\mathsf{C}}$ denotes the summation weights (as convex combination, so $\sum_{\mathbf{w}} \mathbf{w} = 1$) for any parent-child pair. C is simply a placeholder to denote a child node and C must be either a leaf, sum or product node itself. Regarding the SPN output $\mathcal{S}$, it is computed at the root node (that is, the evaluation of $\mathbf{x}$ is the post-propagation result at the root of the DAG and we have $\mathcal{S}(\boldsymbol{\lambda}) = \mathcal{S}(\mathbf{x})$) and the probability density for $\mathbf{x}$ is then given by $p(\mathbf{x}) = \frac{\mathcal{S}(\mathbf{x})}{\sum_{\mathbf{x}' \in \mathcal{X}} \mathcal{S}(\mathbf{x}')}$. They are members of the family of probabilistic circuits (Van den Broeck et al., 2019). A special class, to be precise, that satisfies properties known as completeness and decomposability. Let N simply denote a node in SPN $\mathcal{S}$, then

$$\mathbf{sc}(\mathsf{N}) = \begin{cases} \{X\} & \text{if N is an indicator node} \\ \bigcup_{\mathsf{C} \in \mathrm{ch}(\mathsf{N})} \mathbf{sc}(\mathsf{C}) & \text{else} \end{cases} \tag{4}$$

is called the **sc**ope of N and

$$\forall \mathsf{S} \in \mathcal{S} : (\forall \mathsf{C}_1, \mathsf{C}_2 \in \mathrm{ch}(\mathsf{S}) : \mathbf{sc}(\mathsf{C}_1) = \mathbf{sc}(\mathsf{C}_2)) \tag{5}$$

$$\forall \mathsf{P} \in \mathcal{S} : (\forall \mathsf{C}_1, \mathsf{C}_2 \in \mathrm{ch}(\mathsf{S}) : \mathsf{C}_1 \neq \mathsf{C}_2 \implies \mathbf{sc}(\mathsf{C}_1) \cap \mathbf{sc}(\mathsf{C}_2) = \varnothing) \tag{6}$$

are the completeness (i) and decomposability (ii) properties respectively. While the two properties are crucial for what is meant by a SPN, they are not of interest for any further consideration in this work. Nonetheless, to briefly mention the intuition: (i) is a property on the sum nodes which enforces that any children pair for any given sum node needs to have the same scope and (ii) is a property on product nodes which enforces that any children pair for any given product node must have disjoint scopes (unless we consider duplicate nodes). Both conditions (i) and (ii) together make an SPN have the well-known tractability properties, for example if one were to consider marginal inference, that is, computing the integral $\int p(\mathbf{x})\mathrm{d}\mathbf{x}$, then (i) will allow to "push down" integrals onto children ($\int p(\mathbf{x})\mathrm{d}\mathbf{x} = \sum_i w_i \int p_i(\mathbf{x})\mathrm{d}\mathbf{x}$ for each child $i$), whereas (ii) will allow to decompose complicated integrals into more manageable ones ($\int\int p(\mathbf{x}, \mathbf{y})\mathrm{d}\mathbf{x}\mathrm{d}\mathbf{y} = \int p(\mathbf{x})\mathrm{d}\mathbf{x} \int p(\mathbf{y})\mathrm{d}\mathbf{y}$). Since their introduction, SPNs have been heavily studied such as by (Trapp et al., 2019) that present a way to learn SPNs in a Bayesian realm whereas (Kalra et al., 2018) learn SPNs in an online setting. Several different types of SPNs have also been studied such as Random SPN (Peharz et al., 2020b), Credal SPNs (Levray & Belle, 2020) and Sum-Product-Quotient Networks (Sharir & Shashua, 2018)) to name a few. More recently, on the intersection of machine learning and causality, Zečević et al. (2021a) proposed an extension to the conditional SPN (CSPN, Shao et al. (2019)) capable of adhering to interventional queries. Formally, an iSPN is being defined as

$$\mathcal{I} := (f, \mathcal{S}) \quad \text{s.t.} \quad \boldsymbol{\psi} = f(G_{\mathcal{M}}) \quad \text{and} \quad \mathrm{Dom}(\mathcal{S}) = \mathbf{V}_{\mathcal{M}} \tag{7}$$

for some SCM $\mathcal{M}$, that need not be known (only its graph and arbitrary many interventional data sets), where $\mathbf{V}_{\mathcal{M}}$ are $\mathcal{M}$'s endogenous variables (all variables with 'names') and $G_{\mathcal{M}}$ is the causal graph implied by $\mathcal{M}$. Meaning that $\boldsymbol{\psi}$ provides parameters to the SPN $\mathcal{S}$ by knowing which causal graph to consider, whereas $\mathcal{S}$ generates distributions for variables $\mathbf{V}_{\mathcal{M}}$ based on parameters $\boldsymbol{\psi}$ (note: we drop the notation for both $f$ and $\mathcal{S}$ to reduce clutter, $f := f_{\boldsymbol{\theta}}, \mathcal{S} := \mathcal{S}_{\boldsymbol{\psi}}$ i.e., communication between the two sub models of $\mathcal{I}$ happens via $\boldsymbol{\psi}$). In this sense, the iSPN is a special case to the standard CSPN formulation proposed by (Shao et al., 2019) where the neural net parameterizer $f$ and the generative SPN $\mathcal{S}$ are chosen in a particular way to correspond to some SCM $\mathcal{M}$. That is, consider the general formulation of a CSPN $\mathcal{C} = (f, \mathcal{S})$ modelling a conditional distribution $p(\mathbf{Y} \mid \mathbf{X})$ with feed-forward neural network $f : \mathbf{X} \mapsto \Psi$ and SPN $\mathcal{S} : \mathbf{Y} \mapsto [0, 1]$. By realizing that an intervention $do(\mathbf{x})$ comes with the mutilation of the causal graph $G = (V, E)$ such that new graph is $G' = (V, \{(i, j) : (i, j) \in E \land i \notin \mathrm{Pa}_i\})$, the iSPN is able to formulate an intervention for SPN natural to the occurrence of interventions in structural causal model. The neural net parameterizer (sometimes also referred to as gate model) $f$ orchestrates the *do*-queries such that the density estimator (SPN) can easily switch between different interventional distributions. An alternate approach to causality but also through the lens of tractability was recently considered by (Darwiche, 2021).

These computational models (SPN) oppose the classical notion of semantic models (e.g. BNs, see Fig.2), they trade off interpretability with efficiency. That is, an SPN might be difficult to decipher, similar to other neural methods like the multi-layer perceptron (MLP), but offer even *linear* time inference—while the BN (like the Pearlian SCM) directly reasons about qualitative relationships, similar to a finger-pointing child, but at *exponential* cost.

## 3 Are There Even Different Types of Causal Models?

Before we can realize our discussion of the important matter around tractability of inference, we need to take a step back and look at causal model (families) themselves. That is, it is important for us to understand that causal models themselves are not a matter of choosing between black or white (metaphorically, for being causal or not) but rather live on a continuous spectrum that defines different extents to which a causal model can be considered 'causal.' Only if we start differentiating between these different models, it will become apparent how they naturally exhibit different inference procedures and help making the case as suggested by the title that "*not all causal inference is the same.*"

If we consider the literature in AI/ML research before the introduction of SCMs by Pearl (2009), we typically only find tangential references to causality through concepts such as 'invariance' or 'manipulation.' Many concepts such as counterfactuals have been used in robotics and planning literature prior to Pearl without the actual naming or any direct reference to the idea of causality (even in a broader sense beyond Pearl's

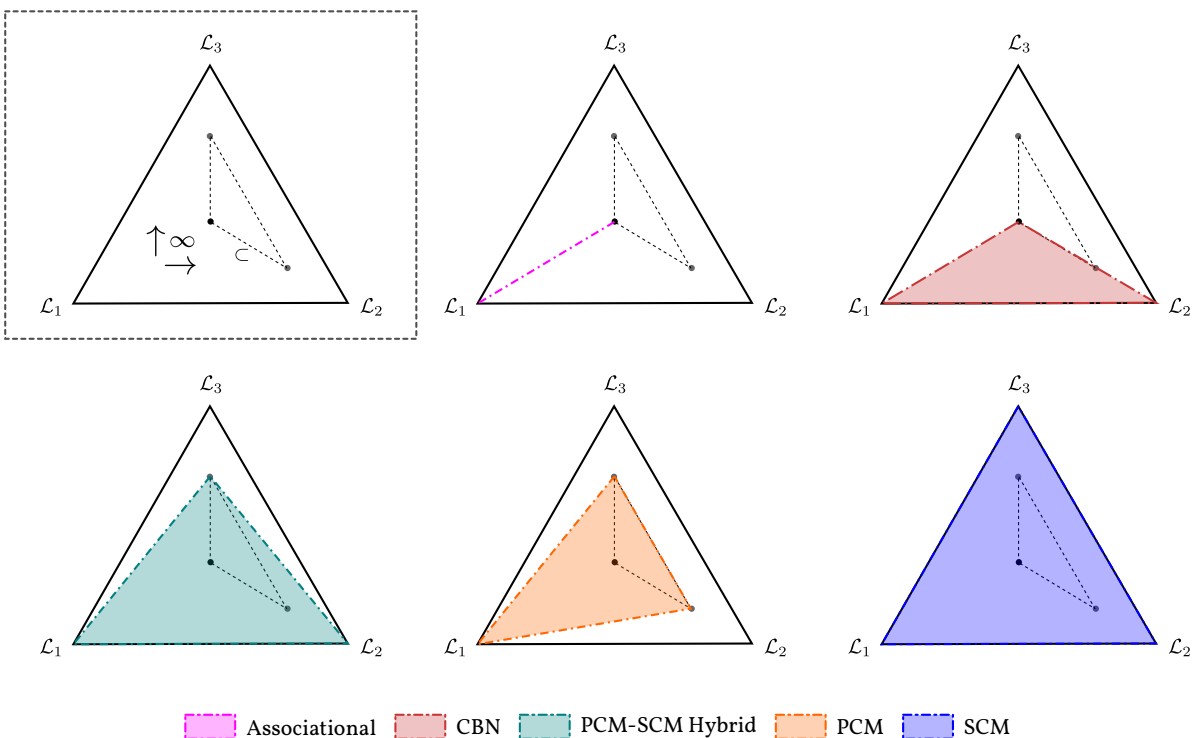

Figure 1: **Classifying Causal Models based on PCH Expressivity.** Legend (top left): $\mathcal{L}_i$ denote the PCH levels, mid-point denotes $\mathcal{L}(\text{model}) \subset \mathcal{L}_i$, whereas outer-point denotes $\mathcal{L}(\text{model}) = \mathcal{L}_i$. Partial Causal Models are those models that span an area that is neither minimal (Associational) nor maximal (SCM). Note how the classical CBN falls into the category of PCM. (Best viewed in color.)

formalism). Fortunately, the AI/ML literature has been free of stigmata regarding the 'C-word', opposed to the long controversies in the medical sciences and econometrics (see Hernán (2018)), lending itself to a rather quick adaption. Nonetheless, most of the literature ever since the introduction of SCMs has treated the property of a model being 'causal' as binary, cf. Kaddour et al. (2022) which discuss various new models that lie at the intersection of causality and classical ML approaches. In our first contribution with this paper we propose to think of causal models in a more diverse manner. Not just the inference processes are different (as we will see in subsequent sections), but the models themselves. Actually, the former is a necessary consequence of the latter. Similar to the analysis of various intricacies of the SCM family by Bongers et al. (2021) (cf. Fig.7 giving an overview on "causal graphical models"), we make the suggestion that one can classify different types of causal models using their expressivity with respect to the PCH levels. We illustrate this classification in Fig.1 by visualizing the three levels (associational, interventional and counterfactual) on a simplex and then drawing areas, for the different models, within these simplices to illustrate their expressivity. Since $\mathcal{L}_2, \mathcal{L}_3$ are uncountably infinite, the outer-point of those levels are reserved to generative models like the SCM that can generate infinitely many such distributions. The mid-point naturally illustrates that a given model only *partially* expresses this level, thus we give it the the name of *partially causal models* (PCM). We see that the SCM spans the maximal area (the full simplex), whereas any associational model (which includes classical models like variational autoencoder) will simply form a line to $\mathcal{L}_1$. Thus, we propose that PCM are indeed all those models that lie in between. Interestingly, this class of models is richer than one would initially suspect, we examine this now.

Arguably the most interesting case is the historic one i.e., the *Causal Bayesian Network*. The CBN was introduced before the SCM and is also a 'causal' model, however, it never stood on the same rung as the SCM since it is incapable of defining and therefore expressing counterfactual quantities. This further gives justification to the synonym of SCM, namely Functional BNs since they expose the structural equations which make possible counterfactuals (and also a formal notion of hidden confounders). Next, is the class of

*finite* PCMs, that is, models that can provide interventional and/or counterfactual distributions but never generate infinitely many (thus finite). Recent models that fall into this class include the Causal VAE (Yang et al., 2020) or the interventional SPN (Zečević et al., 2021a) (why we think this should be the case and how our definition captures this will be presented in the following). Finally, we recognize a third class of *hybrid* models (denoted "PCM-SCM Hybrid" in Fig.1) which generate both a finite and infinite set of distributions for the causal levels of the PCH but never for both simultaneously (like SCM). For this third class, to the best of our knowledge no models have been proposed yet. Whether such models can be constructed remains to be investigated/proven, however, in the conceptualization presented they take a clear position.

While a proper formal coverage of the idea presented in this section deserves attention in a separate line of work, we want to close off this discussion by providing a more formal, first definition of PCM based on the model classification in terms of PCH levels:

**Definition 1** (Partially Causal Model)**.** *Let $\mathcal{M}$ be a model capable of generating distributions that fall into the distinct levels of the PCH ($\mathcal{L}_i$ for $i \in \{1, 2, 3\}$) and let $\mathcal{L}_i(\mathcal{M})$ denote said set of level $i$ distributions. Further, let $\mathcal{M}^*$ denote the ground truth (or reference) SCM with $\mathcal{L}_i(\mathcal{M}^*)$ like before. If the conditions (i) $\mathcal{L}_1(\mathcal{M}) \neq \varnothing$, (ii) $\mathcal{L}_2(\mathcal{M}) \times \mathcal{L}_3(\mathcal{M}) \subseteq \mathcal{L}_2(\mathcal{M}^*) \times \mathcal{L}_3(\mathcal{M}^*)$, and (iii) at most one causal level $|\mathcal{L}_i(\mathcal{M})| = \infty$ are met, then $\mathcal{M}$ is called partially causal (w.r.t. $\mathcal{M}^*$).*

Let us briefly consider each of the formal aspects presented in the above definition.

**Meaning of $\mathcal{L}$ operator:** First off the Pearlian causal hierarchy and its $\mathcal{L}$evels. As before, the operator $\mathcal{L}_i : \mathbb{M} \mapsto \mathbb{X}$ is a mapping from generative models (e.g. neural net, sum-product net etc. are all examples of potential choices $\mathcal{M} \in \mathbb{M}$) to their generated distributions (e.g. a latent space capable of producing images of cats and dogs $\mathcal{X} \in \mathbb{X}$). Therefore, $\mathcal{L}(\mathcal{M})$ is a set of distributions generated by some model $\mathcal{M}$ and this set is possibly infinite. Finally, the $i$ in $\mathcal{L}_i$ brings in the knowledge on causality by labelling all the distributions within said set according to some level on the causal hierarchy being associational ($i = 1$), interventional ($i = 2$) and counterfactual ($i = 3$). To answer the question where this 'label' comes from, it is basically an implicit assumption that there exists some SCM $\mathcal{M}_{\mathrm{SCM}}$ which is our underlying data generating process (therefore, we often times just omit to mention this underlying ground truth model i.e., we simply define $\mathcal{L}_{1-3} := \mathcal{L}_{1-3}(\mathcal{M}_{\mathrm{SCM}})$). Since $\mathcal{L}_{1-3}(\mathcal{M}_{\mathrm{SCM}})$ are well defined, we simply check whether the corresponding $\mathcal{L}_i(\mathcal{M}_{\mathrm{Learned}})$ for some model of interest $\mathcal{M}_{\mathrm{Learned}}$ match. Unfortunately, since $\mathcal{M}_{\mathrm{SCM}}$ is generally (or typically) unknown, there is no way to check this correspondence in practice. However, with consideration of the existence of such ground truth model $\mathcal{M}_{\mathrm{SCM}}$ (which is reasonable and standard practice in causality research), we can state a sensible definition as done above.

**Meaning of bound on cardinality of $\mathcal{L}_{>1}(\cdot)$:** The presented definition should capture models that lie between 'non-causal' and structural causal models which requires two conditions (a) that they can generate some causal quantities (as measured by some ground truth SCM $\mathcal{M}_{\mathrm{SCM}}$), therefore either $\mathcal{L}_2(\mathcal{M}_{\mathrm{Learned}})$ or $\mathcal{L}_3(\mathcal{M}_{\mathrm{Learned}})$ must be defined/non-empty since only interventional/counterfactual distributions are considered causal (this part (a) therefore corresponds to (ii) in the definition and the following part (b) then to (iii)), and (b) either one of the levels is not allowed to be infinite because we can prove by contradiction that if there was a PCM that was not an SCM but has both $|\mathcal{L}_2(\mathcal{M}_{\mathrm{Learned}})| = \infty$ and $|\mathcal{L}_3(\mathcal{M}_{\mathrm{Learned}})| = \infty$, then there exists some SCM $\mathcal{M}'$ which matches both levels, that is, $\mathcal{L}_2(\mathcal{M}_{\mathrm{Learned}}) = \mathcal{L}_2(\mathcal{M}')$ and $\mathcal{L}_3(\mathcal{M}_{\mathrm{Learned}}) = \mathcal{L}_3(\mathcal{M}')$, therefore being indistinguishable from the given SCM, which is a contradiction to the premise that $\mathcal{M}_{\mathrm{Learned}}$ is not an SCM. Put differently, this proof by contradiction also gives meaning to what we mean by 'partial' from the other end of the spectrum i.e., while (a) lower bounded our 'causalness' to be non-zero, (b) upper bounded it to not be fully causal like SCM, thus being right in-between somewhere as a partially causal model.

While we will not make any further use of this definition subsequently, it is still worth pointing out how the PCMs in Sec.5 all abide by the above definition. To revisit our PCM examples from earlier, both the CausalVAE and the iSPN can be considered as PCM following our definition. If we trust our definition to capture the essence of what it means to be 'partially' causal, which we made sure is the case through the thorough discussion in the previous paragraphs, then satisfying the criteria in our definition is both necessary and sufficient for a model to be considered a PCM. Next, we will exemplify this with the iSPN model. The iSPN, further denoted as $\mathcal{V}$, is a generative model based on a neural net parameterizer and a generative

SPN that jointly allow for modelling both observational and interventional distributions (as suggested by the i in the name of iSPN). Therefore, any iSPN $\mathcal{V}$ models $\mathcal{L}_1(\mathcal{V})$ and $\mathcal{L}_2(\mathcal{V})$ with $|\mathcal{L}_2(\mathcal{V})| = N$ where $N$ is the number of experimental distributions available for training, according to the regime discussed in the original paper. Since counterfactuals cannot be covered, $\mathcal{L}_3(\mathcal{V})$ is undefined. Thus $\mathcal{V}$ satisfies conditions (i-iii) and is considered a PCM. An analogue analysis can be performed for the CausalVAE, as it is also capable of generating interventional distributions but not counterfactual ones, therefore, it could never be an SCM since $\mathcal{L}_3$ distributions are a necessary condition. Hypothetically speaking, however, if the CausalVAE on the other hand were able to generate $\mathcal{L}_3$ distributions, then it might be considered as an SCM. This would not be the case if one allowed the same for the iSPN (i.e., to generate $\mathcal{L}_3$ distributions) since the iSPN can only produce causal distributions up to the training data available, whereas any model that can *identify* causal distributions from observational data and other assumptions like the causal graph (like the CausalVAE) would automatically fall out of the PCM into the SCM class of models.

To briefly mention the following structure of the paper. In this section we answered the question whether different causal model existed in the first place affirmatively by providing a first classification in visual terms and then a formal definition of PCM. In the next three sections (Secs.4,5,6) we will discuss the three key causal model families (non-causal, partially causal and structurally causal) in terms of *inference*. Then we summarize our insights in terms of a taxonomy in Sec.7 and subsequently use it to propose a new model (as addenum to the main discussion) in Sec.7.1.

## 4 Inference in Non-Causal (or Correlation-Based) Models

To expand further on the boundaries of the integration between causality and machine learning, we first perform an inspection on how causal inference can occur with correlation-based models. Fig.2 schematizes the basic, "naïve" approach to classical causal inference that we investigate in this section. One takes the *do*-calculus to perform the actual "causal" inference, and then takes the available observational data and a model of choice (e.g. NN/MLP, SPN, BN) to acquire the actual estimate of the query of interest. More specifically, we will focus on SPN from the previous section, since they come with guarantees regarding probabilistic reasoning (opposed to e.g. MLPs) and guarantees regarding their inference tractability (opposed to e.g. BNs). This investigation is important since assuming the wrong causal structure or ignoring it altogether could be fatal w.r.t. any form of generalization out of data support as suggested in (Peters et al., 2017). Central to said (assumed) causality is the concept of intervention. Although being a wrong statement as suggested by results on identifiability, the famous motto of Peter Holland and Don Rubin *"No causation without manipulation"* (Holland, 1986) phrases interventions as the core concept in causality. In agreement with this view that distributional changes present in the data due to experimental circumstances need be accounted for, we focus our analysis on queries $Q = p(\mathbf{y} \mid do(\mathbf{x}))$ with $(\mathbf{x}, \mathbf{y}) \in \text{Val}(\mathbf{X}) \times \text{Val}(\mathbf{Y}), \mathbf{X}, \mathbf{Y} \subset \mathbf{V}$ respectively. $Q$ lies on the second (interventional) level $\mathcal{L}_2$ of the PCH (Pearl & Mackenzie, 2018; Bareinboim et al., 2020).

We first define the concept of a statistical estimand ($\mathcal{L}_1$) for SPN as the application of the rules of probability theory (and Bayes Theorem) to the induced joint distribution.

**Definition 2.** *An SPN-estimand is any aggregation in terms of sums or products of conditionals, $p(\mathbf{x}|\mathbf{y})$, and marginals, $p(\mathbf{x})$, where $p(\mathbf{v}) \in \mathcal{L}_1$ is the joint distribution of SPN $\mathcal{S}$ and $\mathbf{X}, \mathbf{Y} \subset \mathbf{V}$ respectively.*

Before continuing with our insights on SPN-estimands, a short detour on an important assumption. In the most general case, we require a positive support assumption to ensure that an arbitrary SPN-estimand will actually be estimable from data. This is a simple consequence of computing probabilistic ratios $P/Q$ where $P, Q$ represent probabilities and the operation $P/0$ generally being undefined. More formally, the assumption is given as:

**Assumption 1** (Support). *Let $p(\mathbf{v})$ represent an SPN's modelled joint distribution. Then for any probabilistic quantity $p(\mathbf{x} \mid \mathbf{y})$ (with $\mathbf{Y}$ possibly empty) derivable via the laws of probability and Bayes' rule from $p(\mathbf{v})$ it holds that $p(\mathbf{x} \mid \mathbf{y}) > 0$.*

While the above assumption is arguably a standard assumption of probabilistic modelling, it is nonetheless of practical relevance since we might not always be able to ensure that it holds i.e., often times we might

only be given data that covers a subset and not the full set of possible instantiations of the covariates. This certainly amounts to a drawback of purely 'correlation-based' considerations for estimating causal quantities and will become apparent in the subsequent section. On a comparitive note, this assumption is closely aligned with the common 'positivity' assumption in causal inference. That is, in causal inference we are often times interested in the causal effect of a given treatment on some outcome of interest (e.g. a drug on a patient's disease) and one critical assumption commonly employed for estimating this effect is said 'positivity' assumption where we require that all treatments of interest be observed in every patient subgroup (which is similar to Assumption 1). Returning to SPN-estimands, our first insight guarantees us that we can do causal inference with SPN-estimands as depicted in Fig.2.

**Proposition 1.** *Let $Q \in \mathcal{L}_2$ be an identifiable query (that is, $Q$ can be purely written in $\mathcal{L}_1$ terms). There exists an SPN-estimand that answers $Q$.* ∎

*Proof.* Under Assumption 1, trivially follows from SPN being universal density approximators (Poon & Domingos, 2011) and $Q$ being identifiable. □

Since SPN will act as our estimation model, it turns out that any interventional query derived from a Markovian SCM can be modelled in terms of statistical terms represented by the SCM. Due to hidden confounding, this guarantee does not hold in semi-Markovian models. Prop.1 ultimately suggests that inter-layer inference from $\mathcal{L}_1$ to $\mathcal{L}_2$ remains intact when choosing SPN as means of parameterization. A simple but important realization thereof is that the *do*-calculus (Pearl, 2009) can be used as the identification tool for SPN-based causal inference. While unsurprising from a causal viewpoint, from the perspective of tractable models research the result in Prop.1 provides a new incentive for research on the integration of both fields. A possible explanation for this surprising observation is the skeptical view by Papantonis & Belle (2020). They considered the usage of the SPN-BN compilation method from (Zhao et al., 2015) for causal inference within SPN that failed due to the resulting BN being a bipartite graph in which the variables of interest were not connected (connectivity being crucial to non-trivial causal inference). Before investigating this issue of (in)tractability of causal inference, let's define formally what we mean by *tractable* inference.

**Definition 3.** *Let $R$ denote the variable in which the model's runtime scales (e.g. the number of edges in the DAG for an SPN, the number of variables in the DAG for a BN). A scaling of $\mathcal{O}(poly(R))$ of polynomial time is called tractable.*

Note that *poly* includes high-degree polynomials (e.g. $x^{3004}$) and that for SPN we usually have $poly := x$, that is, linear time complexity. It is also important to note that $R$ is different for different models, but interestingly, the number of edges for SPNs does not "explode" exponentially—so indeed, SPNs are far more efficient computation-wise even in practice. To reap initial rewards, we now prove that causal inference with SPN is in fact tractable.

**Corollary 1.** *Let $Q \in \mathcal{L}_2$ be an identifiable query, $|Q|$ be its number of aggregating terms in $\mathcal{L}_1$ and $R$ be the number of edges in the DAG of SPN $\mathcal{S}$. If $|Q| < R$, then $Q$ is tractable.* ∎

*Proof.* From (Poon & Domingos, 2011) we have that $\mathcal{S}$ does a single term, bottom-up computation linearly in $R$. Through Prop.1 and $|Q| < R$ it follows that $\mathcal{O}(R)$. □

Opposed to (Causal) BN where inference is generally intractable (#P complexity), Cor.1 suggests that any estimand can be computed efficiently using SPN even if the estimand identifies an interventional quantity, thereby transferring tractability of inference also to causal inference.

## 5 Inference in Partially Causal Models

An important restriction of SPN-based causal inference is that the joint distribution $p(\mathbf{v})$ of SPN $\mathcal{S}$ optimizes *all* possibly derivable distributions, thereby diminishing single distribution *expressivity*. That is, how "easily" and how precisely $\mathcal{S}$ can in fact approximate our underlying distribution with $p(\mathbf{v})$. Returning to causal inference, we observe that any causal inference will hold but actual estimation from data will suffer in quality as a consequence thereof. In addition, violations of the positive support assumption might render some

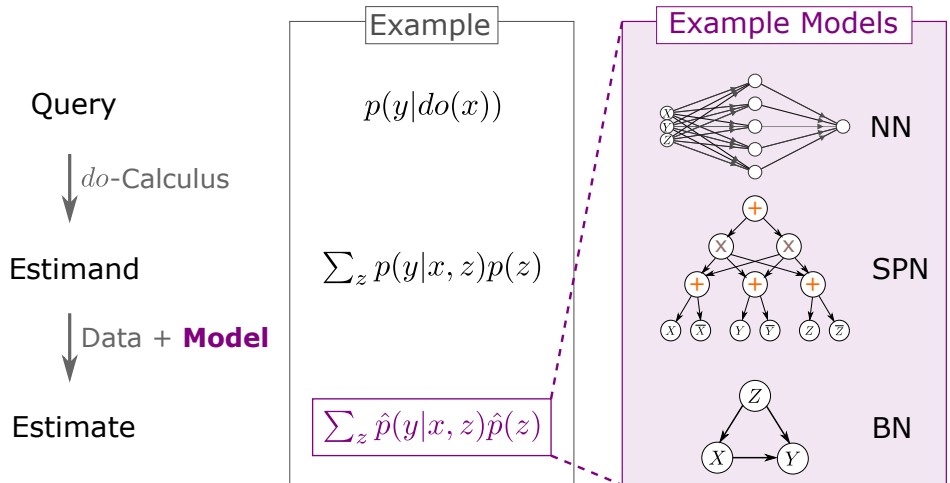

Figure 2: **"Naïve" Causal Inference Schematic.** For any causal *query* we could use the *do*-calculus (Pearl, 2009) to identify a statistical *estimand* (grey), for which there exists a model-based *estimate* (purple). Various model choices are available e.g. a NN, an SPN or a BN. (Best viewed in color.)

practical inference *undefined*. Therefore, in the following we extend our analysis to *partially causal models*, models that actually extend the SPN model class with capabilities from causal inference. More specifically, we consider interventional SPN (iSPN) firstly introduced by (Zečević et al., 2021a). Our first observation is that the iSPN allows for a compressed model description over the SCM, while trading in expressivity since the iSPN has no means of computing the highest level of the PCH $\mathcal{L}_3$ being counterfactuals. The iSPN (Eq.7) is more "powerful" than the SPN by construction, we state formally.

**Proposition 2.** *Let E denote an SPN-estimand (Def.2). There exists a graph G for which the SPN-estimand of iSPN evaluated at G is E.*

*Proof.* There exist always an SCM $\mathcal{M}$ with induced graph $G$ such that the observational distribution of $\mathcal{M}$ and SPN-estimand $E$ correspond accordingly. Since iSPN extend regular SPN via arbitrary causal graphs, simply select $G$ as the graph of choice. $\qquad\square$

Prop.2 further suggests that iSPN are also joint density estimators, although being defined as a special case of conditional estimators (CSPN), and that any SPN will be covered by the observational distribution ($\mathcal{L}_1$) of a corresponding iSPN. In the following, assuming corresponding data $\mathbf{D}_i \sim p_i{\in}\mathcal{L}_2$, we prove that iSPN allow for direct causal estimation of the interventional query ($\mathcal{L}_2$). This sets iSPN apart from the setting in the previous section with regular SPN that were dependent on a "causal inference engine" such as Pearl's *do*-calculus. To illustrate the difference between the approaches, consider the following example,

$$\mathcal{M} := (\{f_X(Z, U_X), f_Y(X, Z, U_Y), f_Z(U_Z)\}, p(\mathbf{U})),$$

and we try inferring the query $Q$ defined as

$$p(y|\, do(x)) = \textstyle\sum_z p(y|x,z)p(z),$$

where the identification equality is given by the backdoor-adjustment formula on $\mathcal{M}$ (Pearl, 2009; Peters et al., 2017). The l.h.s. will be modelled by an iSPN, while the r.h.s. consisting of multiple terms will be modelled by the SPN and *required* the backdoor-adjusment formula.

An important consequence of modelling the l.h.s. is that the shortcomings of single distribution expressivity and positive support are being resolved. To elaborate: the l.h.s. expresses a single distribution, whereas the r.h.s. expresses a combination (through means of products, quotients, sums etc.) of distributions, and although they are *equal* in terms of the content they describe (i.e., the actual values / probability) they

are *unequal* in terms of implementation. An SPN models a joint distribution and thereby any derivable 'sub'-distribution, thus in consequence, modelling a single distribution like on the l.h.s. is more difficult for an SPN than for e.g. an iSPN. Furthermore, since there are no ratios involved in computing the l.h.s. which could otherwise render the computation undefined, Assumption 1 from the previous section does not apply to partially causal models like e.g. the iSPN. What might come as a surprise is that, although we overcome previous shortcomings and dependencies, we do not loose tractability. We observe:

**Proposition 3. (TCI with iSPN.)** *Let $\{Q_i\} \in \mathcal{L}_2$ be a set of queries with $i \in I \subset \mathbb{N}$ and $R$ (as in Cor.1) for iSPN $\mathcal{I}$. Any $\{Q_k\}_k$ with $k \in K \subseteq I$ is tractable.*

*Proof.* There is two cases to consider (1) any fixed $Q_i$ and (2) when we switch between different $\{Q_i\}_i$ for $i \in \mathbb{N}$. For (1), since any iSPN reduces to an SPN upon parameter-evaluation, we can apply Cor.1 and thereby have that $\mathcal{O}(R)$. For (2), we know the iSPN (Zečević et al., 2021a) uses a neural net, therefore we have $\mathcal{O}(poly(R))$. $\square$

Prop.3 seems to suggest that iSPN should always be the first choice as they don't seem to compromise while being superior to the the non-causal models that rely on causal inference engines, however, the iSPN from (Zečević et al., 2021a) comes with strict assumptions. Specifically, we don't assume a causal inference engine but we *require interventional data*—which in a lot of practical settings is too restrictive. Also, in the case of semi-Markovian models, the iSPN falls short. Further, we observe a "switching" restriction, which the non-causal models did not have. That is, when we have to consider multiple interventional distributions the cost will scale w.r.t. to the size of the gate model (either quadratically or cubically for standard feed-forward neural networks).

## 6 Inference in (Parameterized) Structural Causal Models

In the previous sections we discussed non-causal and partially causal models (based on SPN), showing that they are tractable—mostly trading on certain aspects such as assumptions on the data and how many queries we ask for. Although they have tractable causal inference, these methods actually lack in terms of *causal expressivity*. All our previous observations were restricted to queries from $\mathcal{L}_2$, that is, interventional distributions. Why not counterfactuals? Since these models are not *Structural Causal Models*. The Pearlian SCM extended historically the notion of Causal BNs (CBN) by providing both modelling capabilities for *counterfactuals* but also *hidden confounders*.

Now, in the following, we will move onto this more general class of models that is fully expressive in terms of the PCH. For this, consider a recent stride in neural-causal based methods ignited by the theoretical findings in (Xia et al., 2021), where the authors introduced a *parameterized* SCM. Since this term was never coined in the general sense, we provide it here for convenience.

**Definition 4.** *Let $\mathcal{M} = \langle \mathbf{U}, \mathbf{V}, \mathcal{F}, P(\mathbf{U}) \rangle$ be an SCM. We call $\mathcal{M}$ parameterized if for any $f \in \mathcal{F}$ we have that $f(\cdot, \theta)$ where $\theta$ are model parameters.*

In the case of (Xia et al., 2021), the $f$ were chosen to be neural nets. Note that Def.4 actually allows for different function approximators (e.g. a combination of neural nets and other models), however, so far in the literature we usually have the model class $\mathcal{F}$ be only of one such model choice (e.g. $\mathcal{F}$ being the class of MLP, therefore, each $f$ is simply a standard feed-forward net). It is further important to note that any *parameterized* SCM is in fact an SCM—so, an NCM is a valid SCM, furthermore, it implies the complete PCH.

Since SCMs extended CBNs, and since CBNs are not computation graphs (like an SPN is) but rather semantic graphs, we might conclude that SCMs inherit properties of the CBNs when it comes to inference. Unfortunately, it turns out, this heritage of a parameterized SCM is valid and leads to their intractability for causal (marginal) inference.

**Theorem 1.** *Causal (marginal) inference in parameterized SCM is NP-hard.*

*Proof.* The herein presented proof can be considered as a natural (maybe even simple) consequence of long standing results in computation theory by means of (Cooper, 1990). It has been long understood that exact inference in BNs is computationally hard and since SCM historically derive from BN, the presented result might not come as a surprise. Nevertheless, we present a more concrete proof in the following and try to stress that this result is important after all since there has been arguably a disconnect between computation theory and research around causality. Put differently, the importance of SCM for causality opposed to BN is is what makes the Theorem important in the view of the authors of this manuscript. As a foreword to the conceptual portion of the proof and especially readers unfamiliar with computational complexity theory, Cooper made use of a proof technique referred to as *reduction.* That is, problem A is being reduced to problem B by showing that a solution to B will solve A. Naturally, this transfers the complexity i.e., if B is a difficult problem, then successfully reducing A to B shows that A is also a difficult problem. The phrase "A is at least as hard as B" is often times used to convey said idea. More specifically for this result, we will use a 3-SAT reduction where 3-SAT is the problem of evaluating whether the Boolean expression of multiple clauses with 3 literals (a clause, typically written in paranthesis, consists of multiple literals connected by Boolean operators, an example is shown in the following) is satisfiabile or not, for example, is there a combination of $x_{1...6}$ s.t. $B := (x_1 \vee x_2 \vee x_3) \wedge (x_4 \vee x_5 \vee x_6)$ evalutes to true ($B = 1$)? Since it has been established by Cook (1971) that 3-SAT (satisfiability of 3-literal-clause expressions) is NP-hard, we can deduce that our problem A is also NP-hard if we can reduce it to 3-SAT s.t. solving A requires solving 3-SAT. This is being done in the following. The simple proof might only involve reasoning that SCM extend BN such that any inferred solution for any given causal query holds in a corresponding BN if and only if it holds in the SCM because then we simply apply (Cooper, 1990) which showed that BN inference is NP-hard, thus also SCM inference. To write down the full argument, we apply the same technique of 3-SAT reduction. In the first step, we require a mapping between clauses from 3-SAT and SCM. As a reminder, in 3-SAT we talk of literals $Q_1, ..., Q_n$ for $n \in \mathbb{N}$ with $Q_i \in \{0, 1\}$ and clauses of 3 literals each $C_1, ..., C_m$ for $m \in \mathbb{N}$ with $C_i(Q_k, Q_j, Q_l)$ for $k, j, l \in \{1, ..., n\}$. The goal is then to find a configuration of literals $(q_1, ..., q_n)$ such that each clause in the set of all clauses $C = \{C_i\}_{i=1}^m$ evaluates to 1 (read, "true"). We know since (Cook, 1971) that 3-SAT is NP-hard. So it suffices to show a reduction from 3-SAT to SCM inference i.e., that SCM inference is "at least as hard" as 3-SAT or that an oracle of the latter would subsequently solve the former. Let $\mathcal{M} = \langle \mathbf{U}, \mathbf{V}, \mathcal{F}, P(\mathbf{U}) \rangle$ be our SCM. We do the following mapping: (1) each literal $Q_i$ will be in $\mathbf{V}$ and only depend on its "nature" term in $\mathbf{U}$, so for each $Q_i \leftarrow f_{Q_i}(U_{Q_i}) = U_i$ where the $U_i = \mathcal{B}(\frac{1}{2})$ are random coin flips (2) each clause $C_i$ will be an effect of its causes $Q_i$, so for each $C_i \leftarrow f_{C_i}(Q_k, Q_j, Q_l, U_{C_i})$ such that $f_{C_i}$ is an indicator function of the clause. Since all clauses in $C$ need be satisfied, we create a reversed, binary tree denoted by $A_i \leftarrow f_{A_i}(\text{Pa}_{A_i}, U_{A_i})$ (where at the leaves we have $\text{Pa}_{A_i} = \{C_a, C_b\}$ for arbitrary two clauses and for internal nodes $\text{Pa}_{A_i} = \{A_{i-1}, C_c\}$ for some arbitrary clause $C_c$). Finally, we have $X \leftarrow f_X(A_{m-2}, U_X)$ (note $m - 2$ since we had $m$ clauses). This completes the mapping, the second step is to show equivalence of $p(X = 1)$ to the satisfaction of $C$. Our construction implies $p(X = 1) \geqslant p(X = 1|C_s)p(C_s|U_s)p(U_s)$, where $U_s$ denotes "true" for variables in $U$ satisfying every clause in $C$ and $C_s$ correspondingly, and we have $p(X = 1|C_s) = 1$, $p(C_s|U_s)$, $p(U_s) = (\frac{1}{2})^n$ so $p(X = 1) > 0$ when $C$ is satisfiable. If $C$ is not satisfiable, then there must be a term $p(X = 1|C_q)$ for some $q \in \{1, ..., m\}$ such that $p(X = 1|C_q) = 0$. We have $p(X = 1)$ iff. $C$ is satisfiable. Since 3-SAT is NP-hard, we have that marginal inference in SCMs, that is computing queries of the type $p(\mathbf{X} = \mathbf{x})$ for any $\mathbf{X} \subseteq \mathbf{V}$, is NP-hard as well. Since a causal query involves an intervention (e.g. via Pearl's *do*-operator such as $p(\mathbf{X} = \mathbf{x} \mid do(\mathbf{Y} = \mathbf{y}))$ for some SCM $\mathcal{M}$) but amounts to a regular marginal query in the modified SCM $\mathcal{M}^{do(\mathbf{y})}$ where the structural equations of $\mathbf{Y}$ have been replaced to evaluate to $\mathbf{y}$, we subsequently also have that *causal* marginal inference in SCMs is NP-hard. Finally, since parameterized SCMs are all instances of the general class of SCM, where the structural equations $\mathcal{F}$ and the exogenous distribution $P(\mathbf{U})$ are simply parameterized for sake of implementation on general purpose computers, that is $\mathcal{F} \triangleq \mathcal{F}_{\boldsymbol{\theta}}$ and $P(\mathbf{U}) \triangleq P_{\boldsymbol{\theta}}(\mathbf{U})$ with $\boldsymbol{\theta}$ representing free parameters (e.g. the weights of a neural network), we also have that causal marginal inference in parameterized SCMs is NP-hard which concludes our proof. $\square$

From a computational perspective, the result in Thm.1 is a protest against the original formulation of the SCM in terms of long-term suitability for next generation learning systems. Although being an arguably simple consequence of the BN-heritage of the SCM, still, Thm.1 strongly advises against any efforts of using parameterized SCM for real-world impact. Even if the parameterization comes from powerful approximators like neural nets—causal inference remains notoriously intractable. However, for both the sake of completion

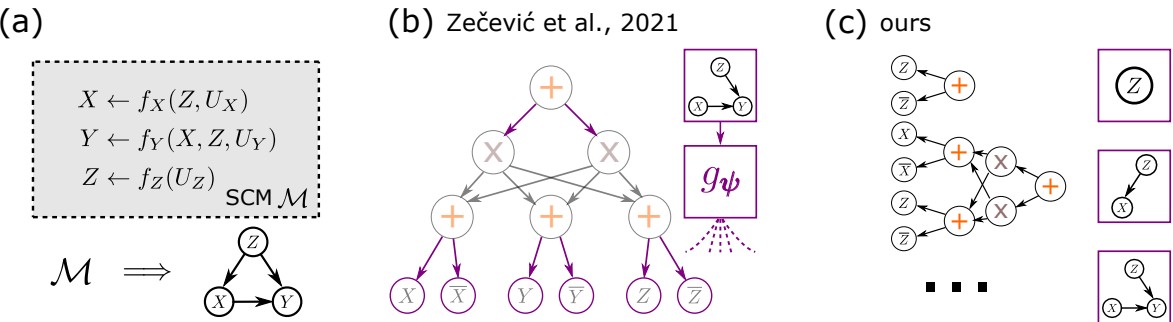

Figure 3: **Tractable Causal Inference Models.** (a) The unobserved SCM $\mathcal{M}$ implies a causal graph $G$ and generates the data to be used for estimation, (b) shows an iSPN (Zečević et al., 2021a) that uses a gate model to estimate causal effects, whereas (c) is a LTNCM (Def.5)—a partially tractable approximation to $\mathcal{M}$. (Best viewed in color.)

and the interest of establishing the theoretical connection in the scope of this systematic investigation, we present for the first time a new parameterization of the NCM using SPN. In spite of Thm.1, this idea is indeed sensible since any *partial* inference within the parameterized SCM might still be efficient. Effectively, the SPN could thereby still offer a more pragmatic alternative to e.g. a neural net since it would not necessarily compromise in terms of predictive performance (model capacity). We believe this to be true since often times the structural equations of an SCM describe reasonably *simple* mechanisms because of their local and thereby somehow restricted nature. Another argument to the same point would be that of causal abstraction (as discussed by for instance (Rubenstein et al., 2017)), which simply means that if our structural equation is not simple, then we can still abstract further into a more fine-grained SCM. That is, an SCM as a collective might be complex, but not its single components—similar to how a neuron in mammalian cortex shows simple activity, but human cognition in total is capable of highly complex decisions. Therefore, we now present the *Linear Time Neural Causal Model* (LTNCM) formally.

**Definition 5** (LTNCM). *Let $\mathcal{M}=\langle\mathbf{U},\mathbf{V},\mathcal{F},P(\mathbf{U})\rangle$ be a parameterized SCM. If $\mathcal{F}$ exclusively defines SPN, then $\mathcal{M}$ is a Linear Time (Mechanism Inference) Neural Causal Model.*

An alternate definition of LTNCM invovles the "General NCM" definition given by Xia et al. (2021), by which the LTNCM is indeed an NCM but with SPN modules instead of classical feed-forward nets. While (General) NCM are still tractable in terms of mechanism inference, they are *polynomial* because of the neural net modules and as we will see simply using SPN modules allows for *linear time* mechanism inference (thus the naming). We also stick the name *neural* over *structural* since SPN can (a) be viewed as a special type of neural/deep model (see Vergari et al. (2019)), and (b) the term "structural" so far seems exclusive to the general formalism of SCM and not to specific (ML) estimators.

Fig.3 provides a schematic comparison of the two causal models based on SPN units i.e., the iSPN (Zečević et al., 2021a) from the section on partially causal models and the LTNCM (Def.5). Evidently, the LTNCM is concerned with a more complex model description (put simply, it requires more models), yet because of that it becomes a causal model fully expressive in terms of the PCH as it poses a subset of the set of all SCMs. On a different note, in Fig.4 we show a visual schematic on the different inference processes that additionally features NCM (Xia et al., 2021). We now state the simple consequence of defining an SCM with SPN units instead of neural nets, which will further reveal one more advantage for preferring SPN over neural nets for parameterized SCM.

**Corollary 2** (NCM versus LTNCM, informal). *Evaluating any structural equation for some SCM $\mathcal{M}$ is non-linear in NCM (Xia et al., 2021) and linear in LTNCM.* ∎

*Proof.* The forward pass for the neural nets (as used by the classic NCM) has polynomial runtime complexity and the concrete polynomial expression will depend on the concrete architecture given. We take this observation as a fact based on results also reported by (Bienstock et al., 2018), where the authors make use of a polyhedral representation to obtain new and better computational complexity results for

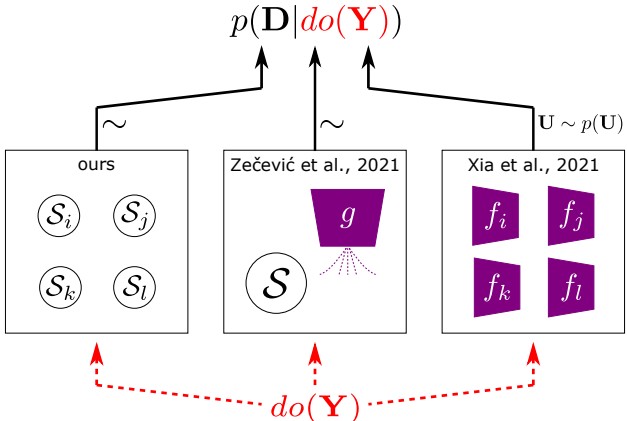

Figure 4: **Schematic Overview of the Inference Processes for Selected Causal Models.** From left to right: LTNCM (Def.5), iSPN (Zečević et al., 2021a), and NCM (Xia et al., 2021). White circles denote SPN, purple quadrangles are NN. (Best viewed in color.)

training problems of well-known neural network architectures with the polyhedron being an encoding scheme for capturing the different neural network training problems that can be considered/parameterized through architecture, activation functions, loss function, and sample-size. An arguably more easily digestible (or casual) reference is following scientific online post by Luna Lux (`https://lunalux.io/computational-complexity-of-neural-networks/`) that describes exactly the forward pass scenario of standard MLP. For regular MLP architectures, the polynomial term is generally lower bounded by a quadratic term. For SPN, as previously established, exact inference time will be linear in the size of the network. □

Cor. 2 suggests that restricted causal inference (e.g. not general marginal inference) even with NCM is tractable but inefficient when compared to LTNCM since the former has at least quadratic runtime complexity opposed to linear for the latter. Said comparison behaves the same for the iSPN, since Prop.3 suggests that (for a fixed iSPN state) any inference will also be linear. Further extending the comparison to other neural-causal models as suggested by (Zečević et al., 2021b), namely NCM-Type 2 and iVGAE. For the NCM-Type 2 we observe (as expected) worse, cubic runtime complexity since modelling occurs on edge-opposed to structural equation level. For the iVGAE (which is a partially causal model), which is comparable to the iSPN in terms of model description, the time complexity is as bad as for the NCM. Therefore, iSPN (Eq.7) offer a clear advantage over other neural-causal models in terms of inference efficiency since any causal query will be answered in linear time, whereas NCM-variants and CBNs have worse time complexities. However, it is important to note that NCM-variants might offer for more expressivity in terms of the PCH.

# 7 Summarizing the Key Differences for Causal Inferences

Conclusively, a researcher might choose one model over the other based on the specific application of interest (e.g. efficacy versus expressivity). Upon investigating these various scenarios for tractable causal inference, we offer a conclusive overview of our tabular taxonomy for inferences in different model families in Tab.1 including neural-causal inferences. Legend: OLS = Ordinary Least Squares, CNN = Convolutional Neural Networks, GAN = Generative Adversarial Networks, FBN = Functional Bayesian Network, iVGAE = interventional Variational Graph Autoencoder (Zečević et al., 2021b), "Causal Circuits" (Darwiche, 2021), CausalGAN (Kocaoglu et al., 2017), NCM (Xia et al., 2021), Deep SCM (Pawlowski et al., 2020).

## 7.1 'Bonus:' An Easy Solution to Speeding Up Mechanism Inference in SCM

After evaluating the taxonomy from Tab. 1, we arrive at the realization that mechanism inference in SCM, while tractable, is still non-linear. Looking closely, we realize that this is due to the neural networks

| Model Family | PCH | Identification | Mechanism Inference | Marginal Inference |
|---|---|---|---|---|
| OLS, CNN, GAN [1] | $\mathcal{L}_1$ | ✗ | - | polynomial |
| SPN [2] | $\mathcal{L}_1$ | ✗ | - | linear (Cor.1) |
| CausalVAE [3], iVGAE [4], CausalGAN [5], CT [9] | $\mathcal{L}_2$ | ✗ | - | polynomial |
| iSPN [6] | $\mathcal{L}_2$ | ✗ | - | linear (Prop.3) |
| CFQP [10], DiffSCM [11] | $\mathcal{L}_3$ | ✗ | - | polynomial |
| NCM [7], DeepSCM [8], CAREFL [12] | $\mathcal{L}_3$ | ✔ | polynomial (Cor.2) | intractable (Thm.1) |
| LTNCM | $\mathcal{L}_3$ | ✔ | linear (Cor.2) | intractable (Thm.1) |

Table 1: **Taxonomy of Inference in Causal Model Families.** *Top*: the three classes perspective of non-causal, partially causal and structural causal models with known models (non-exhaustive list). *Bottom*: Summarizing tractability properties discussed throughout the paper. PCH layer $\mathcal{L}_i$ with $i$ being the upper bound on causal quantities expressible (e.g. $i = 3$ means any causal quantity according to Pearl can be generated). Identification suggests that cross-layer inferences can be performed (e.g. no external identification engine like *do*-calculus is necessary). A dash (-) denotes that the structural equations of a corresponding SCM can not even be defined in the given model family. Marginal inference refers to whether the general computation scheme $p(x) = \sum_{\mathbf{v} \setminus x} p(x, \mathbf{v})$ is computable tractably. Mechanism inference refers to the tractability of the computation of any single sub-module (i.e., structural equation). The original paper references as labelled within the "Model Family" column of the table above: [1] (Goodfellow et al., 2015), [2] (Poon & Domingos, 2011), [3] (Yang et al., 2020), [4] (Zečević et al., 2021b), [5] (Kocaoglu et al., 2017), [6] (Zečević et al., 2021a), [7] (Xia et al., 2021), [8] (Pawlowski et al., 2020), [9] (Melnychuk et al., 2022), [10] (De Brouwer, 2022), [11] (Sanchez & Tsaftaris, 2022), [12] (Khemakhem et al., 2021).

parameterizing the mechanisms. If we look at SPN as base $\mathcal{L}_1$ models, then we see how marginal inference is linear. A simple solution we arrive at by having this quick look at the taxonomy is that SCM parameterized by SPN should lead to linear-time mechanism inference. Indeed, this is the case as we show in this quick empirical illustration.

**Training and Estimation with LTNCM.** Since LTNCM are a special case of SCM with SPN as parameterizing units, we can apply inference in the same way. That is, we make use of the truncated factorization formula (Pearl, 2009) by choosing a sample (or Monte Carlo) based approximation thereof,

$$p(\mathbf{V} = \mathbf{v}|\ do(\mathbf{X} = \mathbf{x})) \approx \frac{1}{m} \sum_i^m \prod_{\mathbf{V} \setminus \mathbf{X}} f(\mathbf{v}, \theta), \qquad (8)$$

where $m$ is the number of samples for the unmodelled/noise terms $U_i$ and $f$ as in Def.4. The intuition behind this formula is that an intervention will mutilate the original causal graph deleting dependence on $\mathbf{X}$'s parents. To perform training, one can simply resort to the maximization of the probability in terms of the negative log-likelihood to account for numerical stability, that is $\theta^* = \arg\min_{\theta \in \Theta} -\frac{1}{n} \sum_i^n \log(p(\mathbf{v}|\ do(x)))$ where $n$ is the number of data points. The consistency criterion refers to the assumption that a query like $p(y = 1, x = 1|\ do(x = 0))$ should automatically evaluate to zero since it would be inconsistent to observe $x$ as opposing the intervention.

We investigate the newly-introduced LTNCM (Def.5) specifically. We first "sanity check" the model by checking for causal effect and general density estimation. Then we conduct two experiments regarding tractability of causal inference. More specifically, we answer the following questions:

**Q1.** To which degree are causal effects being captured on qualitatively different structures?

**Q2.** How is the estimation quality for interventional distribution modelling?

**Q3.** How does time complexity scale when increasing the SCM's size, that is, number of modelling units $(N_U)$?

**Q4.** How does time complexity scale when increasing the size of each unit per SCM structural equation $(S_U)$?

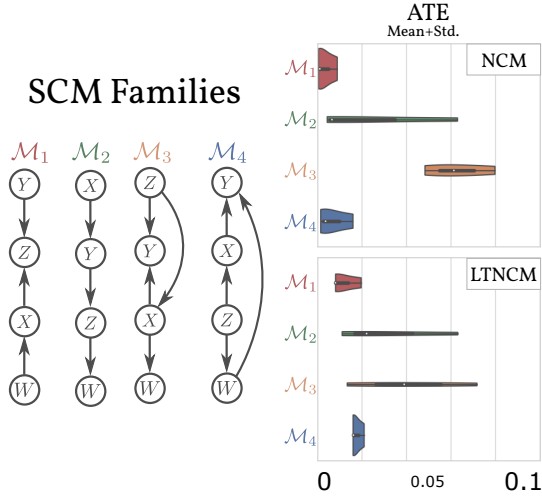

Figure 5: **ATE.** Averaged over multiple random seeds over multiple paramterizations of the given SCM. Both NCM and LTNCM perform well in estimating causal effects. (Best viewed in color.)

|  | SCM | $\mathcal{L}_1$ | $\mathcal{L}_2^{do(X=0)}$ | $\mathcal{L}_2^{do(X=1)}$ |
|---|---|---|---|---|
| NCM | $\mathcal{M}_1$ | .011 | .010 | .026 |
|  | $\mathcal{M}_2$ | .017 | .011 | .020 |
|  | $\mathcal{M}_3$ | .012 | .009 | .030 |
|  | $\mathcal{M}_4$ | .006 | .010 | .005 |
| LTNCM | $\mathcal{M}_1$ | .075 | .040 | .310 |
|  | $\mathcal{M}_2$ | .012 | .032 | .022 |
|  | $\mathcal{M}_3$ | .032 | .024 | .033 |
|  | $\mathcal{M}_4$ | .029 | .021 | .011 |

Table 2: **Density Estimation.** Averaged Jensen-Shannon divergence values on three different distributions for each of the four SCM families and for both NCM and LTNCM.

Questions **Q1-2** serve to validate that the presented model is indeed causal, whereas questions **Q3-4** serve to show the intended speed up in terms of mechanism inference. For details regarding our synthetic data sets (that is, the used SCM families), the overall protocol and hyperparameters—we point to appendix A.1.

*TL;DR.* The questions **Q1-2** are both answered in favor of LTNCM (see Fig.5 and Tab.2), that is, both causal effect estimation as well as general density estimation are competitive with standard NCM, while **Q3-4** confirm our previous discussions in terms of general inference being intractable and mechanism inference being linear for the LTNCM (see Fig.6).

**Q1.** *Causal Effect Estimation.* We observe adequate modelling of the ATEs in both neural-causal models. The worst score on ATE for this binary setting would be 2, while the observed values are in the range $[0, 0.09]$ thus significantly less. The confounded cases ($\mathcal{M}_{3/4}$) are indeed inferred correctly. LTNCM with chosen hyperparameters achieves sligthly worse score than the NCM but with the tendency of reduced variance in the estimates. We argue that the observed variances stem from the choice of SCM parameterizations.

**Q2.** *Density Estimation.* We observe adequate modelling of the different densities (the actual plots for NCM are provided [CLICK HERE, NCM Plots] and for LTNCM are provided [CLICK HERE, LTNCM Plots]) since error rates lie mostly in the low single digit domain. Most notably is the increased variance of the $do(X = 1)$ distribution for LTNCM on $\mathcal{M}_1$. Observing closely, we see that even the other distributions already show less-optimal performance. Since all experiments are conducted with the same, simple architectures, we argue that this non-optimization is explanatory.

**Q3.** *Increasing $N_U$.* Consider Fig.6, Left. We increase the size of the system which is arguably the most common form of scaling and relevant to the development of e.g. complex social networks or for biomedical analysis of complex proteins. As predicted by our intractability result in Thm.1 both NCM (Xia et al., 2021) and LTNCM (Def.5) scale *exponentially*, since they are both parameterized SCM that do not represent computational but rather semantic relations of the variables. The offset difference stems from the specifics of our experimental setup and is negligible.

**Q4.** *Increasing $S_U$.* Consider Fig.6, Right. We increase the size of each of the system's models (each structural equation) with the reasoning that in nature it might occur for a causal relation to be notoriously complex, for instance again in the medical domain the causal mechanism that revolves around risks of smoking as long-standing example (Pearl & Mackenzie, 2018)–although one might argue that at a more

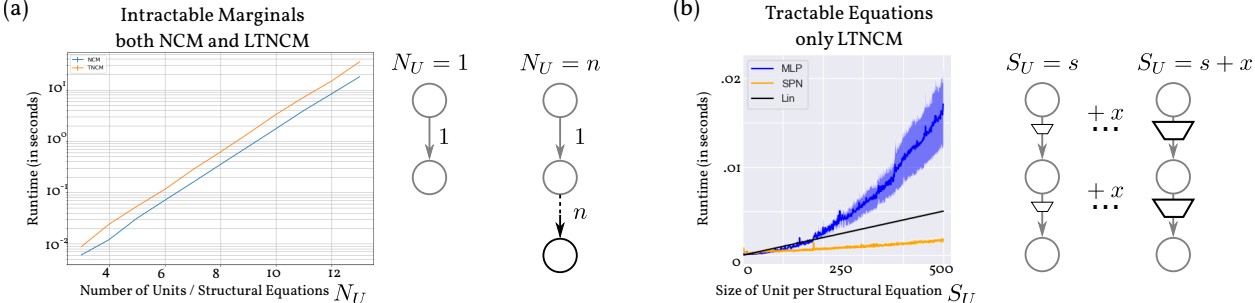

Figure 6: **Experiments Showing Different "Dimensions" of Tractability.** (a) log-scale plot of NCM (Xia et al., 2021) versus LTNCM (Def.5) inference runtime (in seconds) when increasing the size of the SCM i.e., the number of causal variables and thereby both the number of structural equations and subsequent models to be trained. We observe intractability as predicted by Thm.1. (b) regular-scale plot of runtimes for LTNCM and NCM when increasing the capacity of the models to be learned for each of the structural equation. We observe linear tractability only for LTNCM. (Best viewed in color.)

fine-grained view of the mechanisms they might again become simple. As suggested by our simple corollary (Cor.2), only the LTNCM (Def.5) is linear tractable.

**"What the LTNCM can and cannot do."** The LTNCM provides tractable *mechanism* inference. That is, if one wants to evaluate any specific causal effect on a direct link between two variables, then this can be done in linear time. However, arguably most of the time in practice we are interested in general causal effect paths (where we do not know whether the pair we are considering are indeed a cause-effect pair). While one can consider the LTNCM as a clear winner over the classic NCM, it needs to be noted that a further evaluation for use in practice with more demanding benchmarks needs to be consulted before making a definite statement, however, theoretically LTNCM gain the advantage at least on mechanism level. As noted in the beginning parts of this manuscript, however, direct link mechanisms might often times not be too complex to begin with, thus rendering the importance of these gain possibly negligible.

**Concluding Remarks to LTNCM.** As suggested by the title of this subsection, we've considered the presentation of this new LTNCM model as a 'Bonus.' The purpose of the presented experimental primer was to provide a proof of concept for both capturing the differences in causal model inferences as well as actually implementing such different models. Further study of specifically the LTNCM model and its derivatives is purposefully left for study in future work.

## 8 Concluding Discussion and Future Directions

To establish our discussion of causal inference and tractability, we first identified the necessary distinctions between existing models. Our proposed 'spectrum' of causal models, which was classified in terms of expressivity on the PCH, was comprised of "non-causal", "partially causal" and "structural causal" models. Since the middle class has never been discussed in the literature prior to this work, we provided a first technical definition as a foundation for further investigation. We then delved into each of these model families with our discussion of tractable causal inference. We highlighted the importance of tractability for long-term development of practical, next-generation learning systems while providing a broad, comprehensive overview of the discriminative properties for the existing 'zoo' of models (see our summary section). Establishing this overview involved, among other things, that we proved the general intractability result for parameterized SCM (see Thm.1)–which we did using the classical techniques previously used for belief networks. As a 'bonus,' we also showed ways of coping with existing intractability as revealed by our tabular taxonomy. We did so by demonstrating a new model, called LTNCM, which is a NCM that uses SPN-modules to boost mechanism inference to linear time (opposed to polynomials in standard NCM).

We believe that future models will fall into the defined spectrum of causal models and with them their tractability properties. Since future models will require both number-wise more interactions and also more

complex interactions, research at the integration of causality and AI/ML will inevitably encounter the tractability question. Causal inference engines like the *do*-calculus are a tremendously powerful tool, yet ultimately, *complete automation* is what AI seems to aim for. As suggested in Tab.1, making partially causal models like iSPN (Zečević et al., 2021a) "less partial" or structural causal models like NCM (Xia et al., 2021) "more tractable" both aim at the same end result—*tractable causal models*. Coming from the tractability perspective, the original introduction of SPNs from ACs (Darwiche, 2003; Poon & Domingos, 2011)—that allowed for replacing semantic relations through computational ones—might provide hints for a tractable view onto the Pearlian notion of causality. Also, providing a large-scale example akin to ImageNet (Krizhevsky et al., 2012), might be beneficial for future investment into tractable causal inference research.

**Final Remarks.** We hope that our definition of partially causal models, the impossibility result for parameterized SCM, alongside the taxonomy for tractability in causal models and its initial model (the LTNCM) can raise awareness for this novel research direction. We dearly hope so since achieving success with causality in real world downstream tasks will not only depend on learning correct models as we also require having the practical ability to gain access in finite resources to model inferences, ideally as efficiently as possible.

## Acknowledgements

The work was supported by the Hessian Ministry of Higher Education Research, Science and the Arts (HMWK) via the DEPTH group CAUSE of the Hessian Center for AI (hessian.ai). This work was partly funded by the ICT-48 Network of AI Research Excellence Center "TAILOR" (EU Horizon 2020, GA No 952215) and by the Federal Ministry of Education and Research (BMBF; project "PlexPlain", FKZ 01IS19081). It benefited from the Hessian research priority programme LOEWE within the project White-Box, the HMWK cluster project "The Third Wave of AI." and the Collaboration Lab "AI in Construction" (AICO) of the TU Darmstadt and HOCHTIEF.

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

# A  Further Details

We make use of this appendix following the main paper to provide some additional material.

## A.1  Details for the 'Bonus' Experiments on Speeding Up Mechanism Inference in NCM

**Data Sets.** Since we are interested in qualitative behavior in light of the theoretical results established previously, we consider custom SCM simulations. For instance consider the following four families: the collider SCM given by

$$\mathcal{M}_1 = \begin{cases} X \leftarrow & f_X(W, U_X) = W \wedge U_X \\ Y \leftarrow & f_Y(U_Y) = U_Y \\ Z \leftarrow & f_Z(X, Y, U_Z) = X \vee (Y \wedge U_Z) \\ W \leftarrow & f_W(U_W) = U_W \end{cases}$$

and a simple chain SCM which has no confounding given by

$$\mathcal{M}_2 = \begin{cases} X \leftarrow & f_X(U_X) = U_X \\ Y \leftarrow & f_Y(X, U_Y) = X \wedge U_Y \\ Z \leftarrow & f_Z(Y, U_Z) = Y \wedge U_Z \\ W \leftarrow & f_W(Z, U_W) = Z \wedge U_W, \end{cases}$$

and the confounded SCM is given by

$$\mathcal{M}_3 = \begin{cases} X \leftarrow & f_X(Z, U_X) = Z \vee U_X \\ Y \leftarrow & f_Y(X, Z, U_Y) = (X \wedge U_Y) \oplus (Z \wedge U_Y) \\ Z \leftarrow & f_Z(U_Z) = U_Z \\ W \leftarrow & f_W(X, U_W) = X \wedge U_W, \end{cases}$$

and the backdoor SCM given by

$$\mathcal{M}_4 = \begin{cases} X \leftarrow & f_X(Z, U_X) = Z \oplus U_X \\ Y \leftarrow & f_Y(W, X, U_Y) = X \wedge (W \wedge U_Y) \\ Z \leftarrow & f_Z(U_Z) = U_Z \\ W \leftarrow & f_W(Z, U_W) = Z \wedge U_W, \end{cases}$$

where $\oplus, \vee, \wedge$ denote logical XOR, OR, and AND. Note that (for simplicity of analysis) we consider binary variables, however, (LT)NCM naturally extend to the categorical and continuous variables. Note that the collider is an unconfounded structure, thereby conditioning amounts to intervening, $p(y|x) = p(y| do(x))$, while for the backdoor this equality does not hold—thus the causal effect from $X$ on $Y$ is confounded via the backdoor $X \leftarrow ...$ over nodes $Z, W$. We choose $U \sim \text{Unif}(a, b)$ to be uniform random variables each, and we randomize parameters $a, b$.

**Protocol and Parameters.** To account for reproducibility and stability of the presented results, we used learned models for four different random seeds and for each parameterization of any given underlying SCM. For the NCM's neural networks, we deploy simple MLP with three hidden layers of 10 neurons each, and the input-/output-layers are $|\text{Pa}_i| + 1$ and 1 respectively. For the LTNCM's SPNs, we deploy simple two-layer SPNs (following the layerwise principle introduced in Peharz et al. (2020a)) where the first layer consists of leaf nodes, the second layer of product nodes, the third layer of sum nodes and a final product node aggregation. The number of channels is set to 30. We use ADAM (Kingma & Ba, 2014) optimization, and train up to three passes of 10k data points sampled from the observational distribution of any SCM. For experiments in which the size of the SCM is being increased, we use a simple chain and extend it iteratively. For experiments in which the capacity of the mechanism (or units) of the parameterized SCM are being increased, we use a fixed chain SCM structure and scale the model capacity linearly. I.e., the MLPs increase

their hidden layers neurons number while SPNs increase their layer channel. For causal effect estimation, we focus on the average treatment effect given by $ATE(T, E) := \mathbb{E}[E|\, do(T = 1)] - \mathbb{E}[E|\, do(T = 1)]$ that for the binary setting reduces to probabilistic difference $p(Y = 1|\, do(X = 1)) - p(Y = 1|\, do(X = 0)) = ATE(T, E)$. For measuring density estimation quality, we resort to the Jensen-Shannon-Divergence (JSD) with base 2 that is bounded in $[0, 1]$ where 0 indicates identical probability density functions i.e., an optimal match in terms of JSD.

## A.2 Code Repository and Details of Technical Setup

This brief section provides anchor points for any further relevant information.

**Code Repo.** Our code repository alongside visualizations is publically available at: `https://github.com/zecevic-matej/Not-All-Causal-Inference-is-the-Same`

**Technical Details.** All experiments are being performed on a MacBook Pro (13-inch, 2020, Four Thunderbolt 3 ports) laptop running a 2,3 GHz Quad-Core Intel Core i7 CPU with a 16 GB 3733 MHz LPDDR4X RAM on time scales ranging from a few seconds up to an hour for the longest experiment setting.

