# OpenReview forum: "Not All Causal Inference is the Same"
_TMLR — Accepted by TMLR_

### Review · Reviewer_dmSA · 2023-06-26

**Summary Of Contributions:**

The authors delve into the tractability of different types of causal inference, examining the conditions, costs, and feasibility of such inferences. They present an impossibility result for tractable inference within parameterized SCM and introduce the concept of partially causal models, shedding light on the trade-offs between model expressivity and inference tractability. Additionally, they propose LTNCM, a linear time mechanism inference method, as an alternative to the polynomial complexity of standard NCM.

**Audience:**

Yes

**Broader Impact Concerns:**

Not applicable.

**Claims And Evidence:**

Yes

**Requested Changes:**

Please refer to the weaknesses and minor issues above.

**Strengths And Weaknesses:**

### Strengths:
The exploration of partially causal models and the investigation of tractability in causal models make valuable contributions to the research field.


### Weaknesses:
1. The paper is not well-written and hard to follow. Several concepts and notations are introduced unclearly or confusingly. For example, the section on sum-product networks in Section 2 is perplexing. The following points highlight some areas of confusion:

- When ‘IV’ is first introduced in equation (4), it would be helpful to provide the full meaning or expand its acronym to ensure clarity and understanding for readers.

- Please clarify the meaning of $S(\lambda)$ and $C(\lambda)$ in equation (3). Additionally, explain the concept of 'scope' of a node in equation (4). When introducing new concepts, please provide adequate explanations. Please address the role of $P$ in equation (6).

- The meaning of 'iSPN' and equation (7) remains unclear.

2. One area that requires improvement is the inclusion of relevant related works. For instance, the paper should discuss the identifiability of interventional/counterfactual probabilities across more types of models, like the one based on VAE, GAN and normalizing flow.

3. The contribution and novelty of the paper appear limited. The proposed LTNCM method seems to be a specific case of NCM, lacking distinctiveness.

4. In Section 3, during the introduction of 'finite PCMs', it is ambiguous what is meant by 'models that can provide interventional and/or counterfactual distributions but never generate infinitely many.' Specify what is generated. Moreover, in Definition 1, clarify the meaning of the operator 'X' and the absolute value of a certain level of PCH.


### Minor concerns:
- In the third line of the last paragraph in Section 3, 'pf' should be corrected to 'of'.

- In Table 1, it would be better to explicitly show the legend and citation of each model instead of including it within the paragraph.

---

> ### Author Response · Authors · 2023-07-17
> **Answer to Reviewer dmSA**
>
> We thank the reviewer dmSA for the detailed comments/questions/observations which helped us in improving our paper.
>
> Regarding the introduction and overall coverage of SPN (and related topics such as iSPN), we've realized that the writing on this part was bad (especially compared to the rest of the paper) and have fixed this now. The formalism should be clear and the different concepts that are being introduced also described with some intuition. Thanks again for pointing this out, it has improved the introductory section a lot.
>
> Regarding the extension of Tab.1 and the discussion of more causal models: we've done so now and cover a total of 12 different papers that discuss causal models. It might come as a surprise that while there are many works on causal inference and also its integration with ML, there are a lot less works on actual causal models for ML. The total of 12 that we cover now ranges from all the ones you mentioned to some more niche ones and we hope to cover most (if not all) now.
>
> Regarding LTNCM as a special case of NCM: indeed, if we consider the general formulation of NCM and further if we consider SPN as a sort of special instance of neural nets, then one could consider LTNCM as special case of NCM. Even if done so, it is an improvement (at least theoretically, but the initial empirics corroborate on this as well) over the previous model. Nonetheless, consider also the comments of fellow reviewer 6CSs regarding LTNCM, following which we've extended the paper accordingly with necessary discussions.
>
> Regarding the formal discussion of the new PCM class we propose: we've improved the details within the Definition itself to make the formalism more clear/precise. Furthermore, please also consider once again the changes made for fellow reviewer 6CSs, as we've almost completely rewritten the PCM section, improving on all aspects covered by both your and reviewer 6CSs' reviews (color code purple).
>
> Regarding typos and similar: thanks again for pointing this out, all of them are also fixed naturally.
>
> We look forward to some possible future discussions, thanks again!
>
> Your authors
>
> p.s. we've incorporated all the requested changes from your end, **your color code is blue.**

---

### Review · Reviewer_6CSs · 2023-06-29

**Summary Of Contributions:**

This paper explores the tractability of existing causal inference methods and proposes potential approaches that are more manageable. It highlights that not all "causal" models are equal, distinguishing between SCM and partially causal models (PCM) capable of answering causal queries. The authors present a tabular taxonomy that categorizes different model families, including correlation-based, PCM, and SCM, based on their tractability properties. Then, they propose Linear Time Neural Causal Model (LTNCM) in conjunction with SPN, which enables linear time mechanism inference. By leveraging the simplicity of individual components described using SPN, LTNCM achieves efficient causal inference.

**Audience:**

Yes

**Claims And Evidence:**

Yes

**Requested Changes:**

I hope the authors can address the weaknesses mentioned above.

**Strengths And Weaknesses:**

Strengths:

1. This paper offers valuable insights for enhancing the efficiency of existing causal inference methods.
2. The proposed model, LTNCM, demonstrates the capability of performing causal inference with linear time complexity. By integrating SPN modules into NCM, LTNCM leverages the simplicity and efficiency of SPN to achieve efficient causal inference.

Weaknesses:

1. In Section 3, the definition of finite PCMs is not clear. The authors mentioned that finite PCMs are models that can provide interventional and/or counterfactual distributions but never generate infinitely many. Can the author provide an example and discuss the definition of finite PCMs further? Why does CausalVAE belong to this class of model?
2. The definition of $\mathcal{L}_i$ is not clear, especially conditions ii and iii. It would be better to explain them more in detail.
3. For the definition of LTNCM, why does LTNCM need multiple models? what do these models represent?
4. What is the cost of LTNCM to reduce the time complexity of causal inference to linear? Theorem 1 shows that Causal (marginal) inference in parameterized SCM is NP-hard, but Definition 5 and Corollary 2 show that LTNCM could reduce the time complexity of causal inference to linear. This is a counter-intuitive conclusion.

---

> ### Author Response · Authors · 2023-07-17
> **Answer to Reviewer 6CSs**
>
> We thank the reviewer 6CSs for the detailed comments/questions/observations which helped us in improving our paper.
>
> Regarding our newly proposed formalization of PCM we've almost entirely rewritten/improved upon the original coverage by making sure to specify concretely what all the operators mean (and how said meaning is motivated) and also by making use of examples and being more concrete on the iSPN/CausalVAE part. Please consider all the updates for your color code.
>
> Regarding LTNCM and multiple sub-models: the sub-models represent the different structural equations, which is a property of SCM that LTNCM (as special instance of SCM) inherit. An SCM is called a "model" as in "single" model but it is really a set of models (a model each for each of the structural equations).
>
> Regarding an extended LTNCM, as with the requests by the other reviewers, we've added an extended discussion which also clearly highlights the "cost of reducing to time complexity." In summary, it is that we can only do mechanism inference for direct cause-effect pairs and that we cannot be entirely sure that practically LTNCM will always outperform NCM, however, theoretically on mechanism inference properties it does (as we've proven).
>
> We look forward to some possible future discussions, thanks again!
>
> Your authors
>
> p.s. we've incorporated all the requested changes from your end, **your color code is purple.**

---

### Review · Reviewer_KHeR · 2023-07-02

**Summary Of Contributions:**

This paper provides a theoretical perspective to show that neurally-parameterized structural causal models (NCM) cannot return an answer to a query in polynomial time. Moreover, the authors also claim that a model can still answer a causal query even if it is not an SCM, which indicates partially causal models (PCM). By carefully demonstrating the background works and discussing the theoretical properties of PCM, a novel direction of causality is proposed by this paper. Additionally, several experiments are conducted to validate the tractability of NCM and TLNCM.

**Audience:**

Yes

**Broader Impact Concerns:**

There are no ethical implications.

**Claims And Evidence:**

Yes

**Requested Changes:**

- Please polish the paper writing for better demonstration.
- The theoretical analyses are not formal and solid enough for proposing a novel causal direction.
- It is suggested to provide more empirical evidence for both effectiveness and efficiency.

**Strengths And Weaknesses:**

Strengths:
- This paper is novel and might be promising in future research on causality.
- This paper is well-organized and generally reasonable.

Weaknesses:
- The writing of this paper needs to be further polished. Many descriptions and demonstrations are not straightforward, which makes it quite hard to capture the true intention.
- Too much informal theoretical proof. For example, Theorem 1, and Corollary 2, which are two major theoretical contributions of this paper, their derivations lack rigorous mathematical derivation, which makes the results unconvincing. It is suggested to provide formal derivation in future versions.
- Many abbreviations are not carefully introduced, which largely confuses the readers. For example, what are “clauses” and “3-SAT reduction” in Theorem 1?
- It is great to make a bald move to promote causality research, however, the experiments of this paper is quite inadequate, it is suggested to conduct more experiments on both effectiveness and efficiency in future versions.
- Minor issues: Typos: “Before investigating the this tractability of causal inference issue”, “so far in the literature we usually have F be only of one such model choice.”, “since the functions of the structural equations often times describe reasonably simple mechanisms”, “We find that for the NCM-Type 2 worse”, and error symbol “[b!]” in Figure 4 and Table 1.

---

> ### Author Response · Authors · 2023-07-17
> **Answer to Reviewer KHeR**
>
> We thank the reviewer KHeR for the detailed comments/questions/observations which helped us in improving our paper.
>
> Regarding Thm.1 and Cor.2 we've added an extended discussion, also making sure to be more straightforward both about intention and intuition. By mentioning basic ideas like "clauses" and "3-SAT reduction," we realized that we missed to provide some computer science background knowledge on computational complexity theory, which we did now.
>
> Regarding the experiments we've added another paragraph to address the implications of the presented results. As suggested by the title of the subsection, the LTNCM can be considered a "bonus" following our thorough analysis.
>
> Regarding typos/formatting issues, we've fixed all of the ones mentioned by you (and through proof reading some more that went under the radar before).
>
> We look forward to some possible future discussions, thanks again!
>
> Your authors
>
> p.s. we've incorporated all the requested changes from your end, **your color code is orange.**

---

### Author Response · Authors · 2023-07-17
**Revision with Requested Changes Complete**

Dear reviewers,

thank you again for the valuable feedback, as it helped a lot in improving our paper to its new state!

As said, the paper has been revised and the revised paper has now been uploaded (with color and also reviewer-specific color codes).

We hope that you welcome the way we've completed requested changes.

We look forward to some possible future discussions, thanks again!

Best regards,
Your authors

---

### Decision · Action_Editors · 2023-08-12

**Recommendation:** Accept with minor revision

**Comment:**

This paper offers a theoretical vantage point to establish that the feasibility of providing a polynomial-time response to a query is unattainable for neurally-parameterized structural causal models (NCM). Furthermore, the paper asserts the possibility for models beyond the realm of structural causal models (SCM), illustrating the concept of partially causal models (PCM) as a means to address causal queries. The exposition encompasses thorough elucidation of the foundational groundwork and comprehensive examination of the theoretical attributes of PCM, thereby introducing a fresh avenue in the domain of causality.

The reviewers expressed their primary apprehension regarding the presentation of the paper, attributed to the inclusion of technical terminology spanning both causal inference and SAT. The revised version demonstrated substantial enhancements in presentation. Nonetheless, a reviewer still voiced concerns about the clarity of the theoretical segment. Although my review didn't identify any significant issues, I would recommend the authors to refine this section for improved readability. For instance, in the proof of Proposition 1, the authors referenced a straightforward outcome from a prior paper based on the positive support assumption. However, this assumption's explanation is absent in the current paper. It would enhance the self-contained nature of the paper to incorporate a definition for the positive assumption.

**Audience:**

Yes

**Claims And Evidence:**

Yes

---

> ### Author Response · Authors · 2023-09-14
> **Minor Revision is Online**
>
> Dear AE, Dear Reviewers,
>
> thank you once more for the great discussions that improved our paper to its revised version.
>
> For the technical / conceptual portion of the changes that we completed for the minor revision please consider the color coded text passages (highlighted in Blue). As requested, we've added the positive support assumption (i) explaining its meaning, (ii) justifying its use, (iii) formalizing it and (iv) comparing it to the positivity assumption typically done in causal inference. Furthermore, the clarity and the chain of thoughts in the end part of the proof of Theorem 1 have been improved.
>
> The code repository is ready but remains anonymous until a final `green light' from the AE. In the case said green light is granted, we will upload the deanonymized camera-ready version of the paper alongside the public GitHub repo and a short video presentation.
>
> We wish to thank everyone involved for the constructive discussions.
>
> Kind regards, your authors